# ARCHPILOT: A PROXY-GUIDED MULTI-AGENT APPROACH FOR MACHINE LEARNING ENGINEERING

## ABSTRACT

Recent LLM-based agents have demonstrated strong capabilities in automated ML engineering. However, they heavily rely on repeated full training runs to evaluate candidate solutions, resulting in significant computational overhead, limited scalability to large search spaces, and slow iteration cycles. To address these challenges, we introduce ArchPilot, a multi-agent system that integrates architecture generation, proxy-based evaluation, and adaptive search into a unified framework. ArchPilot consists of three specialized agents: an *orchestration agent* that coordinates the search process using a Monte Carlo Tree Search (MCTS)-inspired novel algorithm with a restart mechanism and manages memory of previous candidates; a *generation agent* that iteratively generates, improves, and debugs candidate architectures; and an *evaluation agent* that executes proxy training runs, generates and optimizes proxy functions, and aggregates the proxy scores into a fidelity-aware performance metric. This multi-agent collaboration allows ArchPilot to prioritize high-potential candidates with minimal reliance on expensive full training runs, facilitating efficient ML engineering under limited budgets. Experiments on MLE-Bench demonstrate that ArchPilot outperforms SOTA baselines such as AIDE and ML-Master, validating the effectiveness of our multi-agent system.

## 1 INTRODUCTION

Automating the design of machine learning (ML) pipelines has long been a central objective in AutoML and neural architecture search (NAS), offering the potential to liberate practitioners from the laborious task of manually tuning architectures, training procedures, and optimizing hyperparameters (He et al., 2021). Despite impressive progress in neural architecture search, hyperparameter optimization, and end-to-end AutoML systems, most prior work still operates on constrained optimization space (Ying et al., 2019; Dong & Yang, 2020), which severely limits scalability in real-world settings.

Recently, large language models (LLMs) have demonstrated remarkable capability in code generation, reasoning, and multi-step problem solving. This has led to the emergence of LLM-based ML agents that attempt to autonomously design, debug, and iteratively improve ML pipelines. Early systems such as OpenHands (Wang et al., 2024) and MLAgentBench (Huang et al., 2023) established benchmarks and toolchains for evaluating ML agents, while more sophisticated approaches such as AIDE (Jiang et al., 2025), R&D-Agent (Yang et al., 2025), and ML-Master (Liu et al., 2025) have shown that LLMs can iteratively refine code by combining exploration and reasoning over a tree of candidate solutions. These systems typically generate runnable scripts, execute them to obtain feedback, and use that feedback to improve subsequent generations. However, a major limitation remains: candidate evaluation is still dominated by repeated full training runs, which are computationally expensive and make it infeasible to explore the vast solution space under realistic compute budgets. As a result, these systems either evaluate very few candidates—hurting coverage and diversity—or consume prohibitively large amounts of compute, reducing their practicality in production and research environments.

In this work, we introduce ArchPilot, a multi-agent framework for cost-efficient NAS that explicitly decouples generation, evaluation, and orchestration into three collaborating agents. Unlike prior systems that couple all functionality into a single LLM loop, ArchPilot employs:

- an **Orchestration Agent** (OA) that maintains a Monte Carlo Tree Search (MCTS)-style search tree, selects nodes for expansion using an Upper Confidence Bound for Trees (UCT) criterion (Kocsis & Szepesvári, 2006), backpropagates rewards, and manages structured memory to provide context to GA while preventing redundant exploration,
- a **Generation Agent** (GA) that produces initial drafts, repairs failing pipelines, and proposes atomic, testable improvements conditioned on task description and memory, and
- an **Evaluation Agent** (EA) that instruments and launches proxy-training or full-training scripts, computes multiple proxy signals, adaptively reweights them via ridge-regularized least squares with hard-zero constraints, and maintains a calibrated mapping from proxies to true scores

This separation of concerns allows each agent to focus on its specialty: GA maximizes diversity and code quality, EA provides principled and adaptive evaluation, and OA allocates compute and ensures global search consistency.

A key novelty of ArchPilot is its *multi-proxy evaluation with adaptive reweighting*. Rather than relying on a single heuristic or costly full training, EA evaluates each candidate using a small set of cheap but complementary proxies—such as one-epoch validation, noisy validation, and feature-dropout validation (Li et al., 2020; Ha et al., 2023)—to capture generalization, robustness, and feature reliance. It then forms a direction-aligned weighted sum of these signals to obtain a node value for tree search. As labeled tuples $(y, \boldsymbol{x})$ accumulate from full-training runs, EA refits the proxy weights and OA triggers a *tree restart*, rescoring and reseeding from the top-$k$ verified nodes to keep exploration aligned with the most up-to-date evaluation signals.

Overall, ArchPilot turns NAS into a closed-loop, reasoning-driven process that balances efficiency and accuracy: GA continuously proposes diverse candidates, EA provides fast yet adaptive evaluation, and OA strategically allocates expensive full-training budget where it is most informative. This design enables ArchPilot to explore a significantly larger portion of the search space under the same compute budget compared to prior LLM-based systems, while maintaining reproducibility and interpretability through structured memory and node-level statistics.

Our contributions can be summarized as follows:

- We present ArchPilot, a three-agent NAS framework that cleanly separates code generation, proxy-based evaluation, and tree-based orchestration, enabling modular upgrades and fine-grained control over search dynamics.
- We introduce a principled multi-proxy evaluation pipeline with direction alignment, normalized aggregation, and ridge-regularized weight fitting under a hard-zero policy, yielding reliable value estimates even under noisy or partially missing signals.
- We develop a restart-enabled MCTS search algorithm that refreshes exploration whenever the scoring semantics change, preventing stale statistics from misleading the search and focusing compute on empirically strong regions.

We evaluate ArchPilot on MLE-Bench (Chan et al., 2024), demonstrating consistent improvements over strong baselines such as AIDE (Jiang et al., 2025) and ML-Master (Liu et al., 2025). For example, ArchPilot surpasses 66% of teams on average on MLE-Bench Lite, while the number is 60% for ML-Master and only 55% for AIDE. Our results show that adaptive proxy optimization and restart-enabled exploration together yield higher true performance at lower compute cost, underscoring the promise of multi-agent and reasoning-driven NAS.

## 2 RELATED WORK

### 2.1 TRADITIONAL NAS

Neural architecture search (NAS) aims to automatically discover high-performing neural networks, but early approaches were prohibitively expensive because each candidate architecture required full training and evaluation (He et al., 2021). This motivated a variety of strategies to approximate candidate performance more efficiently. One common approach is to restrict the search space and apply surrogate models, e.g., early-stopping, weight-sharing, or training on reduced datasets (Pham et al., 2018; Liu et al., 2018). EcoNAS and related methods further improved efficiency by adaptively determining early-stopping thresholds to terminate poor candidates sooner (Zhou et al., 2020).

Figure 1: Overview of ArchPilot. The Orchestration Agent (OA) selects candidate nodes using MCTS, maintains memory, and coordinates the search process. The selected node, together with its context, is passed to the Generation Agent (GA), which drafts, debugs, or improves training scripts. The Evaluation Agent (EA) then executes proxy training or full training, producing proxy vectors, aggregated scores, and optional true metrics.

Although effective, these approaches still require gradient updates for each candidate, limiting scalability under tight compute budgets.

To further reduce cost, researchers developed *zero-cost proxies*—metrics that can be computed without any training, often from a single minibatch or even a single forward/backward pass (Mellor et al., 2021). Representative examples include SNIP and GRASP, which prune weights based on sensitivity measures (Lee et al., 2018; Wang et al., 2020), and SynFlow, which computes gradient flow in a data-agnostic manner (Tanaka et al., 2020). These methods are fast and dataset-agnostic, but their correlation with true accuracy can vary across tasks and architectures. Recent work such as NAS-Bench-Suite-Zero systematically benchmarked over a dozen zero-cost proxies across 28 tasks and found that combining multiple proxies improves Kendall's Tau correlation with ground-truth performance by up to 42% (Krishnakumar et al., 2022). This finding motivates the multi-proxy aggregation and weight optimization used in ArchPilot, which dynamically reweights or prunes proxies based on their predictive fidelity.

## 2.2 LLM-DRIVEN ML AGENTS

With the rise of LLMs, there is growing interest in using them as autonomous ML engineers. Open-Hands provides a unified platform for running ML tasks with tool-augmented LLMs and defines a standardized evaluation protocol (Wang et al., 2024). MLAgentBench extends this by evaluating agents on machine learning experimentation tasks across data preprocessing, model training, and evaluation (Huang et al., 2023). AIDE frames ML engineering as a code optimization problem over a tree-structured solution space and uses LLMs to iteratively generate, debug, and improve code solutions (Jiang et al., 2025). R&D-Agent goes beyond code refinement by performing literature review, hypothesis generation, and automated experimentation, targeting a full research–development loop (Yang et al., 2025). Most relevant to our work, ML-Master integrates balanced multi-trajectory exploration with steerable reasoning and an adaptive memory mechanism to systematically refine candidate solutions (Liu et al., 2025). However, these systems still rely heavily on full training for evaluation, making them expensive and limiting scalability under realistic resource constraints.

## 3 METHOD

### 3.1 SYSTEM OVERVIEW

We introduce ArchPilot, a multi-agent system for cost-efficient neural architecture search (NAS) that replaces expensive end-to-end training with a learned, adaptive proxy pipeline. The overview of our pipeline is shown in Figure 1. ArchPilot decomposes the problem into three subproblems, each handled by a collaborating agent: (i) a *Orchestration Agent* that coordinates search with an MCTS-style controller, maintains memory, enforces budget, and restarts policies. Below we detail each component and the end-to-end search procedure; (ii) a *Generation Agent* that drafts, debugs,

and improves training scripts and model code; and (iii) a *Evaluation Agent* that curates, executes, and optimizes proxy evaluators and decides when to escalate to full training.

## 3.2 ORCHESTRATION AGENT

The Orchestration Agent (OA) coordinates the end-to-end loop: it (1) maintains *short-term memory* (recent drafts, fixes, proxy/true scores, execution traces) and retrieves *long-term memory* (historical bests and successful nodes); (2) runs a Monte Carlo Tree Search (MCTS) controller with decaying exploration; (3) enforces *time/GPU budgets* at proxy and full-training granularity; and (4) triggers *weight refits* and *tree restarts* when the proxy aggregator changes.

### 3.2.1 SEARCH ALGORITHM

Similar to prior work (Liu et al., 2025), OA adopts MCTS as the search algorithm. However, we rely on proxy evaluations instead of full training when determining the rewards. Each node of our search state $v$ includes a runnable script $c_v$, a proxy vector $\boldsymbol{x}(c_v)$, an aggregated proxy score $s(c_v)$, an optional true score $y(c_v)$ if full training was run, visit stats $(Q_v, N_v)$, and execution results. The search algorithm includes the following four stages.

**Selection.**  Starting at the root, OA recursively picks the child with the highest UCT (Upper Confidence Bound for Trees) until reaching a leaf or a non-terminal node:

$$\text{UCT}(v) = \frac{Q_v}{N_v} + C \cdot \sqrt{\frac{\ln(N_{\text{parent}})}{N_v}}, \tag{1}$$

where $Q_v$ is the cumulative reward of node $v$ (sum of rewards from all visits), $N_v$ is the number of times node $v$ has been visited, and $N_{\text{parent}}$ is the visit count of $v$'s parent node. The constant $C > 0$ controls the exploration–exploitation trade-off, with larger $C$ values leading to more exploratory behavior. We also mark certain nodes as terminal by the same stopping rules as prior work (Liu et al., 2025): (i) if the number of improved steps that fail to exceed a threshold $t$ is greater than $\tau_{\text{improve}}$—and (ii) if consecutive debug attempts exceed $\tau_{\text{debug}}$. This ensures that we are not stagnated in nodes that are hard to improve or debug.

**Expansion.**  From the selected node, the Orchestration Agent (OA) invokes the Generation Agent (GA) to perform various actions, which will be described in detail in the GA section. When calling GA, OA provides a rich context that includes the task description, data signature (input shape, label space, metric direction), available compute resources and package environment, current budget state, and relevant memory.

**Verification.**  Once a new child node is generated and passes basic forward-pass checks, the Orchestration Agent (OA) delegates its evaluation to the Evaluation Agent (EA). EA launches proxy training for the selected node and computes the aggregated proxy score $s(c_v)$. After receiving feedback from EA (proxy scores and, if available, true scores), OA computes a reward signal $R(v)$ that drives the tree search. We adopt the sparse reward decomposition as prior work (Liu et al., 2025):

$$R(v) = \begin{cases} -1, & \text{if node is invalid;} \\ \mathbb{1}\{\text{valid node}\} + \mathbb{1}\{\text{debug successful}\} + \mathbb{1}\{\text{metric improves}\}, & \text{otherwise.} \end{cases}$$

The "metric improves" term is computed with $s(c_v)$ when only proxy scores are available, and with $y(c_v)$ if true scores of both the parent node and the current node are available. This reward encourages functional code, successful debugging, and performance improvement relative to the search frontier.

**Backpropagation.**  The computed reward $R(v)$ is accumulated along the path from node $v$ to the root. Each ancestor updates its visit count $N$ and cumulative value $Q$ according to the standard MCTS update rule:

$$Q_u \leftarrow Q_u + R(v), \qquad N_u \leftarrow N_u + 1 \quad \forall u \in \text{path}(v, \text{root}),$$

ensuring that future selection decisions incorporate both exploration pressure and the new reward signal. This update maintains consistency of the search tree with the most recent proxy aggregation semantics and full-training feedback.

### 3.2.2 Search Tree Tracking and Restarting

OA maintains a *short-term memory* throughout the search run as a journal of all explored nodes and outcomes. For each node $v$, it records (1) the natural-language plan and code, (2) verification results including proxy vectors $x(c_v)$, aggregated proxy score $s(c_v)$, and optional true metric $y(c_v)$, (3) execution artifacts (logs, errors, runtime, cost), and (4) search statistics ($N_v, Q_v$, debug flags). This memory is used to construct context-aware prompts for GA, with parent and sibling summaries preventing redundant fixes and promoting diverse solutions.

The journal is updated continuously. When proxy weights $\lambda$ are optimized, OA recomputes all $s(c_v)$ and *restarts the tree*, reseeding from the top-$k$ nodes under the new aggregator. New proxies are initialized with zero weight to avoid early bias. Tree restarting also prunes overly deep branches, resets visit counts, and shortens GA prompts to keep exploration focused. Node artifacts remain preserved for reproducibility and post-hoc analysis.

### 3.2.3 Budgeting and Cost Control

OA enforces a global compute budget, decrementing it on every proxy evaluation and full training run and terminating once exhausted. Safety checks (e.g., ratio bounds on metrics) prevent spurious scores from distorting UCT statistics.

When the budget is critically low or wall-clock time falls below two hours, OA disables proxy mode and relies solely on full evaluations. Proxy mode is also turned off if excessive buggy nodes suggest poor proxy-task alignment, ensuring the remaining compute is spent on ground-truth signals. As a fail-safe, if a full evaluation fails to produce a valid submission, OA immediately retries on a fresh candidate. These controls keep the search stable, focused, and compute-efficient even under strict resource limits.

### 3.3 Generation Agent

The Generation Agent (GA) produces runnable training pipelines, repairs failing ones, and applies small, testable improvements. GA operates in three modes:

- **Draft:** Given OA's context (task description, dataset preview, installed packages/resources and remaining time/budget, and short-term memory summaries), GA returns (i) a 3–5 sentence natural-language plan and (ii) a complete, self-contained PyTorch script (data loader/preprocessing, model, training loop, and validation). Drafts must be distinct from prior designs referenced in memory.
- **Improve:** For a bug-free parent, GA proposes exactly one atomic modification (e.g., augmentation/regularization, optimizer/scheduler, width/depth). OA supplies the parent code, execution results, and sibling summaries; GA returns a brief rationale plus a modified script implementing only that change so its effect is measurable.
- **Debug:** For failing nodes, GA uses the buggy code, execution logs, and OA's root-cause summary (e.g., shape/dtype/device errors, missing submission file) to produce a minimal fix while preserving previously verified components.

Together, these three modes enable GA to continuously expand the search frontier while maintaining code quality and interpretability of changes. By producing initial drafts, proposing isolated improvements, and repairing failing nodes, GA ensures that every branch of the search tree represents a runnable and progressively refined candidate. This modular design allows the orchestration agent to reason about progress at a fine granularity and facilitates reproducible ablation studies on the impact of individual design decisions.

### 3.4 Evaluation Agent

The Evaluation Agent (EA) is the centerpiece of ArchPilot. It performs three functions: (1) *script synthesis & launch* for proxy or full training, (2) *weighted aggregation* of proxy signals into a single search value, and (3) *weight & proxy optimization* driven by ground-truth observations.

**Proxy and Full Training.** Given GA's runnable script, EA writes an instrumented variant for *proxy training* and launches it in a sandbox; for *full training* it launches the original script under a fixed budget envelope. The proxy variant preserves data/model code, saves a submission file, trains exactly one epoch on 10% of the training data, executes the current proxy registry, and prints a single JSON line with all proxy scores. Each proxy returns a scalar; on failure it emits a large *sentinel*. Since the proxy training script is modified based on the original training script, the proxy functions themselves could lead to exceptions. To save our debugging efforts on the original training scripts, the node will not be treated as buggy if the sentinel numbers are emitted (which indicates that exceptions happen inside the proxy functions).

EA parses stdout to obtain the proxy vector $\boldsymbol{x}(c) = (x_1(c), \ldots, x_m(c))$; if any score is missing or the JSON is malformed, the node is marked invalid for search ranking in this round. We have three initial proxy functions in ArchPilot, which are commonly used in verifying machine learning models:

- **One-epoch validation:** trains the model for a single epoch on a small subset of the training data and reports the mean validation loss, providing a quick estimate of generalization.
- **Noisy validation:** adds Gaussian noise to the input features during validation to assess the robustness and stability of the model's predictions.
- **Feature-dropout validation:** randomly masks a fraction of input features to evaluate how strongly the model depends on specific features and whether its predictions degrade gracefully under feature corruption.

**Weighted Aggregation.** To transform multiple heterogeneous proxy signals into a single scalar guiding search, EA first aligns all proxies to a common "larger-is-better" convention. Each proxy $i$ has a direction coefficient $d_i \in \{+1, -1\}$ (for example, validation losses use $d_i = -1$ so that lower loss becomes higher score). After direction alignment, EA combines the proxies by computing a normalized weighted sum:

$$s(c) = \frac{\sum_{i=1}^m \lambda_i \, d_i \, x_i(c)}{\sum_{i=1}^m \lambda_i}, \qquad \sum_{i=1}^m \lambda_i = 1, \quad \lambda_i \geq 0. \tag{2}$$

This aggregated score $s(c)$ serves as the node value for MCTS selection and reward computation. The weights $\boldsymbol{\lambda}$ are initialized from a conservative prior—often uniform or slightly biased toward the most stable proxy—to avoid overfitting early in the search when little ground-truth data are available.

**Proxy Optimization.** As search progresses and OA escalates candidates to full training, EA collects labeled tuples $\{(y_j, \boldsymbol{x}_j)\}$ where $y_j$ is the true validation metric and $\boldsymbol{x}_j$ the corresponding proxy vector. Once a minimum number of labeled pairs (we use $k=5$) is accumulated, EA refits the aggregation weights to better match the true performance signal. Specifically, let $Z \in \mathbb{R}^{n \times m}$ be the matrix of direction-aligned proxy vectors and $Y \in \mathbb{R}^n$ the vector of true scores (sign-flipped if necessary so that larger is better). EA solves a ridge-regularized least-squares problem:

$$\widehat{\boldsymbol{\lambda}} = \arg\min_{\boldsymbol{\lambda} \in \mathbb{R}^m} \left\| Z\boldsymbol{\lambda} - Y \right\|_2^2 + \alpha \|\boldsymbol{\lambda}\|_2^2, \tag{3}$$

where the regularization parameter $\alpha > 0$ improves stability under collinearity and prevents overfitting. The resulting unconstrained solution $\widehat{\boldsymbol{\lambda}}$ is not guaranteed to be nonnegative or sum to one, so EA projects it back onto the probability simplex:

$$\boldsymbol{\lambda}^* = \Pi_\Delta(\widehat{\boldsymbol{\lambda}}), \qquad \Delta := \left\{ \boldsymbol{\lambda} \in \mathbb{R}_{\geq 0}^m : \mathbf{1}^\top \boldsymbol{\lambda} = 1 \right\}. \tag{4}$$

Here, $\Pi_\Delta$ denotes the Euclidean projection operator onto the simplex, defined as

$$\Pi_\Delta(\widehat{\boldsymbol{\lambda}}) := \arg\min_{\boldsymbol{\lambda} \in \Delta} \|\boldsymbol{\lambda} - \widehat{\boldsymbol{\lambda}}\|_2^2, \tag{5}$$

which ensures that the final weights are valid mixture coefficients and can be interpreted as contributions of each proxy to the overall score.

To ensure robustness, EA also applies a *hard-zero policy* that removes unreliable proxies from the combination entirely:

$$\lambda_i^* = 0 \quad \text{if} \quad \exists j \text{ s.t. } x_i(c_j) \geq \tau_{\text{invalid}}, \tag{6}$$

where $\tau_{\text{invalid}}$ is the sentinel value indicating a failed proxy evaluation. This effectively restricts the fitting to proxies that have never failed and assigns zero weight to the rest. Newly proposed proxies, generated by an LLM when existing proxies are weak or unstable, are added with initial weight $0.0$ so that they do not affect rankings until enough labeled data accumulate for calibration. If the proxy count exceeds a pre-defined budget, EA drops the proxy with the smallest weight to keep the registry compact and focused.

After refitting, EA compares the new weights $\boldsymbol{\lambda}^*$ with the previous ones; if the change exceeds a small threshold,

$$\left\|\boldsymbol{\lambda}^* - \boldsymbol{\lambda}\right\|_1 \;>\; \varepsilon, \tag{7}$$

it updates $\boldsymbol{\lambda} \leftarrow \boldsymbol{\lambda}^*$ and instructs OA to restart the tree. This restart rescales all existing node values using the updated aggregator, resets visit counts and UCT statistics, and reseeds exploration from the top-$k$ nodes with the highest true performance. This procedure ensures that the search tree remains consistent with the most up-to-date scoring semantics and does not get stuck exploring regions that are no longer promising under the refined value estimate.

## 3.5 DISCUSSION

The three-agent design of ArchPilot —Generation, Evaluation, and Orchestration—forms a closed loop that balances exploration, efficiency, and reliability. GA generates runnable code, applies atomic improvements, and repairs failures to ensure valid nodes. EA converts cheap proxy signals into value estimates, refits weights to stay aligned with ground truth, and manages the proxy set. OA coordinates the search with MCTS, propagates rewards, reseeds from strong nodes when proxy semantics shift, and maintains memory to prevent redundant exploration.

This decomposition offers clear separation of concerns: GA handles code generation, EA quantitative evaluation, and OA decision-making and resource allocation. Modularity enables independent upgrades, such as adding new proxies or improving GA prompting, without changing the overall algorithm. By combining proxy-guided search with selective full-training escalation, ArchPilot reduces compute cost while converging to strong solutions. Structured memory and node statistics make the process reproducible and interpretable, enabling ablations and post-hoc analysis.

## 4 EXPERIMENTS

We empirically evaluate ArchPilot on the MLE-Bench benchmark to measure its ability to produce valid, high-performing solutions under realistic compute constraints. Our experiments demonstrate three key findings: (1) ArchPilot achieves higher valid-submission rates and better average leaderboard rankings compared to ML-Master and AIDE, even under the same strict budget; (2) the advantage of ArchPilot is most pronounced on high-difficulty tasks, where proxy-guided search can effectively explore and refine candidate solutions when the full evaluation for each candidate is expensive; and (3) when varying the budget, ArchPilot maintains a strong lead, indicating that its multi-proxy evaluation and tree restart mechanism enable efficient use of limited training resources. Overall, these results confirm that multi-agent proxy-guided NAS delivers superior performance with substantial computational savings.

## 4.1 EXPERIMENT SETTING

We conduct experiments on MLE-Bench (Chan et al., 2024), a comprehensive benchmark consisting of 75 Kaggle-style machine learning tasks spanning tabular, vision, and NLP modalities. Each task provides a fixed dataset split and public leaderboard metric, allowing for standardized comparison of ML agents across domains and difficulty levels. This diversity makes MLE-Bench particularly well-suited to evaluate the generality, robustness, and efficiency of ArchPilot.

**Evaluation Metrics.** We follow the official MLE-Bench evaluation protocol and report several complementary metrics: (1) *Valid Submission Rate*, the fraction of tasks for which the agent produces a runnable pipeline and a valid leaderboard submission file; (2) *Above-Median Rate*, the proportion of tasks where the agent's public score exceeds the median score across all participants; (3) *Bronze+/Silver+/Gold+ Rate*, the proportion of tasks where the agent achieves at least the corresponding Kaggle medal threshold (bronze, silver, or gold); and (4) *Mean Normalized Ranking*,

which measures the relative leaderboard rank of the agent's submission among all teams (lower is better).

The Bronze+/Silver+/Gold+ metrics can be seen as a discretized version of the ranking score, capturing how frequently the agent produces solutions that reach meaningful quality milestones rather than merely counting valid submissions. Together, these metrics provide a holistic view of reliability (can the agent produce runnable solutions?), competitiveness (how often does it beat the median?), and excellence (how often does it achieve medal-worthy performance?).

**Compute Environment.**  All experiments run on a high-performance cluster with A100-SXM4-40GB GPUs (8 per node) and 124 CPU cores per node. We impose a resource-constrained setting of 2.5 GPU-hours per task, excluding model inference time. We additionally evaluate performance under multiple budget levels to study scalability and efficiency. For all agents, we use `gpt-4.1-2025-04-04` as the backbone LLM.

It is worth noting that our hardware configuration is slightly less powerful than that used in ML-Master (Liu et al., 2025), which reports results on A100-SXM4-80GB GPUs.

## 4.2 Experiment Results

**Overall Performance on MLE-Bench.**  Table 1 summarizes the results of AIDE, ML-Master, and ArchPilot across all 75 MLE-Bench tasks. ArchPilot achieves the best performance across all reported metrics, increasing the valid submission rate from 0.867 for ML-Master to 0.893 and reducing the average normalized rank from 0.6535 to 0.6149. These results indicate that ArchPilot is both more reliable at producing runnable pipelines and more effective at generating competitive solutions within the given compute budget. Notably, the gains in Bronze+ and Gold+ rates show that ArchPilot is more likely to discover high-quality solutions even under tight resource constraints, validating the benefits of proxy-guided candidate selection.

Table 1: Overall metrics of different agents on MLE-Bench. Higher is better except for ranking.

| Agent | Valid Submission($\uparrow$) | Above Median ($\uparrow$) | Bronze+ ($\uparrow$) | Silver+ ($\uparrow$) | Gold+ ($\uparrow$) | Ranking ($\downarrow$) |
|---|---|---|---|---|---|---|
| AIDE | 0.787 | 0.240 | 0.173 | 0.133 | 0.107 | 0.6953 |
| ML-Master | 0.867 | 0.267 | 0.173 | 0.147 | 0.107 | 0.6535 |
| ArchPilot | **0.893** | **0.293** | **0.187** | **0.147** | **0.120** | **0.6149** |

**Performance Across Difficulty Levels.**  We further analyze performance across tasks of varying difficulty levels, as defined by the MLE-Bench organizers. Table 2 shows that ArchPilot achieves the largest improvement on high-difficulty tasks, where the average ranking improves from 0.7262 to 0.6469 compared to ML-Master. This is precisely where efficient exploration matters most, since training a single candidate model is particularly time-consuming in these tasks and poor exploration can rapidly exhaust the budget. On medium- and low-difficulty tasks, ArchPilot remains competitive with ML-Master, demonstrating that proxy-guided search does not sacrifice performance in easier settings. These findings align with our design principle: prioritize promising nodes and reduce wasted training on weak candidates, ultimately leading to better budget utilization across all difficulty levels.

Table 2: Ranking of agents on tasks of different difficulty levels. Lower is better.

| Agent | Low ($\downarrow$) | Medium ($\downarrow$) | High ($\downarrow$) | All ($\downarrow$) |
|---|---|---|---|---|
| AIDE | 0.4553 | 0.8150 | 0.7439 | 0.6953 |
| ML-Master | 0.3931 | 0.7755 | 0.7262 | 0.6535 |
| ArchPilot | **0.3333** | **0.7643** | **0.6496** | **0.6149** |

**Budget Sensitivity Analysis.**  Figure 2 reports mean normalized ranking as a function of GPU hours across different task difficulty levels. We observe that ArchPilot outperforms ML-Master and

AIDE across the entire budget range. The improvement is most striking on high-difficulty tasks (Fig. 2(c)), where training a single candidate is expensive and poor exploration wastes substantial budget. In this setting, ArchPilot 's proxy-guided search is particularly advantageous: by filtering out weak candidates before committing to full training, it achieves better rankings with fewer expensive evaluations. For low- and medium-difficulty tasks (Fig. 2(a)–(b)), the gap is smaller but still noticeable, showing that ArchPilot is more compute-efficient even when training is cheap. AIDE often fails to produce valid submissions under tight budgets, explaining its poor early-stage performance. As the budget increases, all methods converge, but ArchPilot maintains a consistent lead, suggesting that its exploration remains focused on promising regions rather than degenerating into random search.

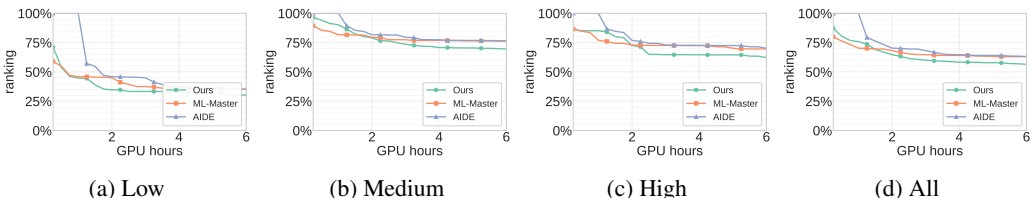

(a) Low       (b) Medium       (c) High       (d) All

Figure 2: Performance vs. GPU budget across difficulty levels. Mean normalized ranking (lower is better) as a function of available GPU hours per task. ArchPilot achieves better (lower) ranking scores across low-, medium-, high-difficulty, and overall tasks.

In summary, these results validate the effectiveness of the three-agent design and the adaptive proxy optimization in ArchPilot. By prioritizing candidates using multi-proxy signals, dynamically reweighting proxies, and restarting the search tree upon significant weight updates, ArchPilot achieves better exploration–exploitation balance, higher reliability, and superior overall performance compared to strong LLM-based baselines.

## 5 DISCUSSION

In this work, we introduced ArchPilot, a multi-agent NAS framework that integrates generation, evaluation, and orchestration agents into a closed-loop system. By leveraging proxy-guided candidate evaluation, adaptive weight fitting, and a restart-enabled MCTS search procedure, ArchPilot significantly reduces the cost of exploring large search spaces while maintaining or improving solution quality. Experiments on MLE-Bench demonstrate that ArchPilot outperforms strong baselines such as ML-Master and AIDE, achieving higher valid submission rates, better rankings, and superior performance under strict GPU budgets.

Despite these promising results, ArchPilot still has limitations. Its performance depends on the availability of informative proxies and on the stability of weight fitting, which may be less reliable in highly noisy or sparse data regimes. Furthermore, while our tree restart mechanism improves search adaptivity, it can occasionally discard promising but underexplored nodes when the scoring rule shifts dramatically. Future work will explore more robust proxy discovery, meta-learned weight initialization across tasks, and principled methods for partial tree reuse to further reduce search overhead.

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
