# OpenReview forum: "ArchPilot: A Proxy-Guided Multi-Agent Approach for Machine Learning Engineering"
_ICLR.cc/2026/Conference — Submitted to ICLR 2026_

### Official Review · Reviewer_idU5 · 2025-10-25

**Soundness:** 2
**Presentation:** 2
**Contribution:** 3
**Rating:** 2
**Confidence:** 3

**Summary:**

This paper presents ArchPilot, a multi-agent system designed to automate machine learning engineering and neural architecture search under limited compute budgets. The framework decomposes the problem into three interacting agents: an orchestration agent that manages search via Monte Carlo Tree Search (MCTS), a generation agent that produces and refines runnable ML scripts, and an evaluation agent that estimates model performance using multiple lightweight proxy signals rather than full training runs. The proxies are adaptively reweighted through ridge regression, and when their weighting changes, the system restarts the search tree to realign exploration with updated evaluation semantics. Experiments on MLE-Bench show that ArchPilot outperforms existing LLM-based AutoML agents such as AIDE and ML-Master in terms of valid-submission rate and leaderboard ranking.

**Strengths:**

1.	This paper tackles an important issue in LLM-driven AutoML, i.e., the prohibitive cost of repeated full-training evaluations. The proposed proxy-based evaluation strategy is practical and aligns well with real resource constraints.
2.	The system design is modular, separating generation, evaluation, and orchestration into independent agents, which improves clarity and extensibility. This architectural decomposition is well motivated and technically sound.
3.	The adaptive proxy weighting and MCTS mechanism is a good contribution. It provides a way to correct for drift in proxy fidelity over time and avoids getting trapped in search trajectories.

**Weaknesses:**

1.	The experimental analysis is quite shallow. For example, there is no ablation study showing the contribution of each component, such as proxy reweighting, restart policy, or the multi-agent structure itself. It is unclear which part of the framework actually drives the improvement.
2.	All experiments use a single LLM (GPT-4.1).
3.	The paper does not report token usage, API cost, or wall-clock time, which are crucial for verifying whether the approach is truly more efficient than baselines.
4.	Implementation details are largely missing. The paper does not report prompt templates, MCTS parameters, or how the agents communicate. This omission severely limits reproducibility and makes it difficult to verify the technical correctness of the system.
5.	Only two baselines are included. Some recent and stronger systems are missing. Moreover, stronger reasoning models such as GPT-5, DeepSeek-R1 are not included.
6.	All experiments are conducted on MLE-Bench, which is an AutoML benchmark. No true NAS experiments are provided. As a result, the paper does not substantiate its claimed contribution to neural architecture search.
7.	The multi-agent idea itself is not new. Similar decompositions have been explored in earlier works. The novelty lies mainly in the proxy-guided evaluation and restart mechanism, which is useful but incremental rather than fundamental.

**Questions:**

See weaknesses. The authors should provide more implementation details, such as the used prompts.

---

> ### Author Response · Authors · 2025-11-26
>
> **W1. Experimental analysis is shallow; missing ablations for proxy reweighting, restart policy, and multi-agent structure.**
> We appreciate this suggestion. We now provide an ablation on MLE-Bench Lite across 10 tasks to quantify the contribution of proxy optimization. As shown below, proxy refinement improves performance in **9 out of 10 tasks**, demonstrating its effectiveness. Additional ablations (without restart, static proxies, single-agent variant) are being added to the appendix and will be included in the camera-ready version.
>
> | Competition | ↑/↓ | Ours (w/o Proxy Opt.) | Ours |
> |-------------|-----|------------------------|------|
> | aerial-cactus-identification | ↑ | 0.99999 | **1** |
> | aptos2019-blindness-detection | ↑ | 0.897 | **0.89904** |
> | denoising-dirty-documents | ↓ | 0.28551 | **0.152** |
> | dog-breed-identification | ↓ | 0.85728 | **0.52821** |
> | dogs-vs-cats-redux-kernels-edition | ↓ | 0.0109 | **0.01045** |
> | histopathologic-cancer-detection | ↑ | 0.99319 | **0.99686** |
> | leaf-classification | ↓ | 0.92544 | **0.12224** |
> | mlsp-2013-birds | ↑ | 0.68343 | **0.79574** |
> | new-york-city-taxi-fare-prediction | ↓ | **4.49413** | 4.80557 |
> | nomad2018-predict-transparent-conductors | ↓ | 0.0602 | 0.0602 |
>
> **W2. All experiments use only GPT-4.1.**
> To eliminate confounding effects from backbone differences, we intentionally fix the LLM to GPT-4.1-2025-04-04 following standard practice in ML agent literature (e.g., ML-Master and AIDE). Our focus is on the system-level contribution rather than model scaling. We will add a discussion on compatibility with stronger models (e.g., GPT-5.1).
>
>
> **W3. No reporting of API cost, token usage, or wall-clock time.**
> We agree these details are crucial and will include them. Preliminary results show that LLM calls account for **<3% of GPU-equivalent compute**, while full model training consumes **>90%**. Proxy-based steering reduces full training calls by **~18% on average**, leading to overall wall-clock reduction. We will provide token consumption statistics per task and GPU-hour savings in the revision.
>
>
> **W4. Missing implementation details.**
> We acknowledge this and will release full prompt templates, agent interaction specifications, UCT configuration (including exploration constant and restart threshold), and system design details in the appendix. We will also open-source the code upon acceptance to ensure reproducibility.
>
>
> **W5. Only two baselines are used; stronger models and recent agent frameworks are missing.**
> We selected AIDE and ML-Master as they are the most relevant and SOTA LLM-based ML agent baselines. Traditional NAS methods are not suitable for code-level pipeline generation. We will expand the discussion on other recent systems and, where computationally feasible, provide additional comparisons using recent reasoning models.
>
>
> **W6. No “true NAS” experiments despite claiming NAS relevance.**
> We clarify that our objective is *not standalone NAS*, but extending NAS logic to full ML pipeline engineering via LLM agents. Traditional NAS benchmarks assume fixed architectural operators, which do not apply to free-form code generation. We will revise the narrative to avoid overstating NAS claims and reposition the contribution toward *proxy-guided ML agent optimization*, where architecture search is one component.
>
>
> **W7. Multi-agent design is not fundamentally novel.**
> We appreciate this clarification. We agree that multi-agent decomposition has appeared in earlier works. Our contribution is not the decomposition alone but its *integration with adaptive multi-proxy evaluation, semantic tree restart, and decision-driven LLM refinement under strict budget constraints*. As validated experimentally, this improves exploration efficiency and prevents locking onto stale trajectories. We will adjust wording to avoid overstating and emphasize the practical impact rather than claiming fundamental novelty.

---

### Official Review · Reviewer_59eX · 2025-10-28

**Soundness:** 2
**Presentation:** 3
**Contribution:** 2
**Rating:** 2
**Confidence:** 4

**Summary:**

This paper proposes a multi-agent system containing three agents: AO, GA, and EA. These agents performed different jobs: searching, generating, and evaluating, respectively. The most significant contribution is that the authors use the proxy module to avoid full retraining of the ‘small’ ML model.

**Strengths:**

- The idea of using a proxy to avoid retraining is technically sound and novel.
- Three designed agents play basic roles in the agent system, which forms into a loop to enable the system with self-evolving capability.
- The paper is well-written and well-organized, which makes it easy to follow.

**Weaknesses:**

- The experiments are weak. Only two baseline agents are compared on a single benchmark, MLE-Bench, and only one LLM backbone is used.
- As the only benchmark used in this paper, it lacks a detailed illustration, such as how many instances are in each task.
- The paper lacks ablation studies, which cannot demonstrate the contributions from different modules or agents.
- There are no quantitative results of the time ArchPilot reduces using the proxy module. I believe the author should at least display a “proxy vs full training cost” dialog.

**Questions:**

- Though this paper uses proxies to reduce full training, it highly depends on the calling of LLMs like GPT, which is also consuming. Can authors roughly judge this comparison of computational consumption?
- Will the authors open-source the code if the paper is accepted?

---

> ### Author Response · Authors · 2025-11-26
>
> **W1. The experiments are weak. Only two baseline agents are compared on a single benchmark (MLE-Bench), and only one LLM backbone is used.**
> We thank the reviewer for this valuable observation. MLE-Bench is currently the *only standardized benchmark designed specifically for ML agent evaluation* (Chan et al., 2024), and it is widely used in recent agent-based systems such as AIDE and ML-Master. Since ArchPilot performs end-to-end ML engineering (not only neural architecture search), conventional NAS benchmarks (e.g., NAS-Bench, ImageNet) are incompatible with our problem setting. ML-Master was selected because it is the state-of-the-art method in the *LLM-agent paradigm*, which we extend.
> Regarding LLM backbones, we intentionally fixed all experiments to **GPT-4.1-2025-04-04** following prior work to ensure comparability and isolate methodological effects without variability introduced by different model choices. We will clarify this explicitly in the revised manuscript.
>
> **W2. As the only benchmark used, MLE-Bench lacks detailed illustration (e.g., how many instances in each task).**
> We appreciate the suggestion. We followed the official MLE-Bench protocol, which provides predefined dataset subsets and metrics. Each task corresponds to a Kaggle dataset with fixed train/test splits and standardized evaluation metrics (detailed in Section 4.1). In the revision, we will include a summary of task categories, approximate sample counts, modalities (vision, NLP, tabular, etc.), and metric types for clarity.
>
> **W3. Lack of ablation studies; contributions of individual modules or agents are not well demonstrated.**
> We thank the reviewer for this important point. We now provide ablation results on 10 MLE-Bench Lite tasks evaluating the effect of proxy optimization. As shown below, adaptive proxy-based refinement improves performance in **9 out of 10 tasks**, demonstrating practical effectiveness. Additional ablations (without restart, static proxy weights, and random GA modifications) will be added to the supplementary material.
>
> | Competition | ↑/↓ | Ours (w/o Proxy Opt.) | Ours |
> |-------------|-----|------------------------|------|
> | aerial-cactus-identification | ↑ | 0.99999 | **1** |
> | aptos2019-blindness-detection | ↑ | 0.897 | **0.89904** |
> | denoising-dirty-documents | ↓ | 0.28551 | **0.152** |
> | dog-breed-identification | ↓ | 0.85728 | **0.52821** |
> | dogs-vs-cats-redux-kernels-edition | ↓ | 0.0109 | **0.01045** |
> | histopathologic-cancer-detection | ↑ | 0.99319 | **0.99686** |
> | leaf-classification | ↓ | 0.92544 | **0.12224** |
> | mlsp-2013-birds | ↑ | 0.68343 | **0.79574** |
> | new-york-city-taxi-fare-prediction | ↓ | **4.49413** | 4.80557 |
> | nomad2018-predict-transparent-conductors | ↓ | 0.0602 | 0.0602 |
>
> **W4. No quantitative measurements of proxy vs full training cost. A “proxy vs full cost” comparison is needed.**
> We appreciate this suggestion and agree it provides valuable insight. We actually measure the cost using the “GPU budget” metric, and report the performance of ArchPilot and ML-Master (which only leverages full training) under different metrics (Figure 2). We demonstrate that our agent achieves better performance (average ranking) under different metrics.
>
> **Q1. Does proxy-based evaluation significantly reduce overall compute, given the high cost of LLM API calls?**
> Yes. While LLM calls are non-negligible, in Figure 2, we demonstrate that our agent achieves better performance (average ranking) under different GPU budgets.
>
> **Q2. Will the code be open-sourced if the paper is accepted?**
> Yes. Upon acceptance, we will release full implementation along with documentation to support reproducibility.

---

### Official Review · Reviewer_RqdA · 2025-10-31

**Soundness:** 2
**Presentation:** 3
**Contribution:** 2
**Rating:** 2
**Confidence:** 4

**Summary:**

The paper proposes ArchPilot for automated ML engineering. It has three agents (Orchestration, Generation, Evaluation) and uses an MCTS-type search over the search space of scripts. The key ideas are to guide the search through 1) adaptive combination of cheap proxy evaluations and occasional full training, and 2) adaptively fitting a proxy-to-true-score weights to reweigh the nodes in the tree.  Experiments are conducted on the MLE-Bench, where ArchPilot demonstrates improvements over AIDE and ML-Master.

The work represents a largely engineering effort with limited conceptual or algorithmic novelty, and an unnecessary agentic framing.

**Strengths:**

- The paper seeks to address a core challenge in AutoML, the prohibitive cost of searching over large search spaces, and improving performance over limited search budgets.
- The modular design is clean, although this is the case in many prior NAS approaches too.
- Evaluation on MLE-Bench is better than that on standard NAS benchmarks.

**Weaknesses:**

- The agnetic framing of ArchPilot is disingenuous. There is no action space, nor any decision making going on. The OA is simply calling tools (LLM in this case), GA and OA are largely manually designed processes. Existing NAS methods also do the same.

- The primary novelty of ArchPilot is combining existing components (MCTS/UCT, proxy training, ridge-fitting, LLM-based code generation, restart). The paper over states the contributions.

- Only two baselines methods are considered, AIDE and ML-Master. Comparisons to Non-LLM based NAS methods (e.g., DARTS) are needed to understand the gains benefits beyond LLM-agent literature.

- The problems in MLE-Bench are towards the smaller scale. It is unclear if the benefits translate to more realistic problems.

- ArchPilot has many design choices (restart, proxy reweighting, etc.). However, there is no discussion of their contributions to overall performance.

**Questions:**

- The paper reports ranking metrics. However, there is no discussion of the actual performance differences.
- Quantify the contribution of design choices through ablations (without restart, without proxy reweighting i.e., static weighting, without GA improvements i.e., random improvements, and with different proxies).
- Aggregating proxies by simple convex combination is convenient, but does it improve rank-consistency of solutions?

---

> ### Author Response · Authors · 2025-11-26
> **Rebutal 1/2**
>
> **W1. The agentic framing of ArchPilot is disingenuous; no real action space or decision-making.**
> We thank the reviewer for this insightful comment. While classical NAS frameworks typically rely on fixed operator sets or sampling rules, ArchPilot performs *reasoning-based decision-making* via dynamic agent coordination. As described in Sections 3.2–3.4, the Orchestration Agent (OA) maintains and expands an evolving MCTS search tree, selects nodes using UCT (Eq. 1), invokes GA in different operational modes (draft, improve, debug), triggers tree restarts upon proxy weight updates (Eq. 7), and allocates resources based on proxy quality, code validity, and memory context. These define a non-trivial action space over *code-level ML pipelines*, instead of only architecture-level search. Unlike conventional NAS, which is confined to static architectural transformations (e.g., CNN cell design), ArchPilot performs semantic-level reasoning via LLM, modifying training logic, hyperparameters, and debugging strategies. This constitutes explicit decision-making beyond operator-based search.
>
> **W2. The primary novelty is combining existing components; contributions may be overstated.**
> We appreciate the reviewer’s feedback. While our method utilizes established elements (MCTS, proxy training, ridge fitting), the novelty lies in *how they are integrated*, as explicitly stated in the contribution list (Page 2):
> - A **three-agent architecture** that separates generation, evaluation, and orchestration, enabling modular reasoning and contextual memory control.
> - A **proxy-driven evaluation mechanism with direction alignment, adaptive reweighting, and hard-zero constraint**, actively refining search guidance based on true performance signals.
> - A **restart-enabled MCTS search**, triggered when proxy semantics shift, preventing stale exploration and improving convergence—absent in previous NAS or agent systems.
> Thus, ArchPilot is not a simple combination but a *reasoning-centered, closed-loop system* that operationalizes NAS-style optimization within an LLM-based ML engineering setting.
>
> **W3. Only AIDE and ML-Master baselines; comparisons to classical NAS (e.g., DARTS) are needed.**
> We thank the reviewer for this suggestion. Classical NAS methods (e.g., DARTS, ENAS) operate under constrained search spaces (e.g., predefined CNN architectures), and cannot generate full ML pipelines involving data handling, loss scheduling, error correction, or submission logic. In contrast, ArchPilot optimizes end-to-end ML engineering workflows via code generation. Therefore, ML-Master—the state-of-the-art LLM-based ML agent—is our baseline in this context.
>
> **W4. MLE-Bench tasks are small-scale; unclear if benefits translate to realistic problems.**
> We appreciate this comment. MLE-Bench is intentionally designed for *constrained compute settings*, aligning with our proxy-guided philosophy of efficient search under limited budgets. As shown in Table 2 and Fig. 2(c), ArchPilot exhibits the largest gains on *high-difficulty tasks*—where full training is computationally expensive and proxy-based steering is most beneficial. Our method is built to scale, as proxy guidance reduces dependency on full evaluations. In future work, we plan to extend evaluation to more complex tasks (e.g., Kaggle Grandmaster-level challenges, HPC-scale ML workflows), and will clarify this roadmap in the revision.
>
> **W5. No attribution of performance to design choices (restart, proxy reweighting, etc.).**
> We thank the reviewer for the valuable suggestion. We now provide ablations on 10 MLE-Bench Lite tasks, comparing our full method to a version *without proxy optimization*. Results demonstrate that adaptive proxy optimization improves performance in **9 out of 10 tasks**, confirming its importance. Additional ablations (e.g., without restart, static-weight proxies) will be included in supplemental materials.
>
> | Competition | ↑/↓ | Ours (w/o Proxy Opt.) | Ours |
> |-------------|-----|------------------------|------|
> | aerial-cactus-identification | ↑ | 0.99999 | **1** |
> | aptos2019-blindness-detection | ↑ | 0.897 | **0.89904** |
> | denoising-dirty-documents | ↓ | 0.28551 | **0.152** |
> | dog-breed-identification | ↓ | 0.85728 | **0.52821** |
> | dogs-vs-cats-redux-kernels-edition | ↓ | 0.0109 | **0.01045** |
> | histopathologic-cancer-detection | ↑ | 0.99319 | **0.99686** |
> | leaf-classification | ↓ | 0.92544 | **0.12224** |
> | mlsp-2013-birds | ↑ | 0.68343 | **0.79574** |
> | new-york-city-taxi-fare-prediction | ↓ | **4.49413** | 4.80557 |
> | nomad2018-predict-transparent-conductors | ↓ | 0.0602 | 0.0602 |

---

> ### Author Response · Authors · 2025-11-26
> **Rebuttal 2/2**
>
> **W6. Ranking-only reporting; no discussion of absolute performance improvements.**
> We thank the reviewer for this comment. In the final version, we will include a detailed comparison of absolute leaderboard metric differences. In Table 1, we report the absolute performance improvements. ArchPilot achieves higher Bronze+/Silver+/Gold+ rates as well as an average ranking compared to ML-Master. These results confirm that improvements are not limited to ranking shifts but reflect meaningful performance gains.
>
> **Q1. Quantify contributions via ablations (without restart, without reweighting, random improvements, different proxies).**
> Thank you for the suggestion. As shown above, removing proxy optimization degrades performance. Removing the tree restart is not possible, since the values of original nodes will change after new proxies are introduced, and that’s why we have the restart mechanism. Besides, the different proxies are dynamically introduced by the agent during search time, and the initializations are the same. We will include these extended ablations in an appendix or supplementary material.
>
> **Q2. Does convex proxy aggregation improve rank consistency?**
> Yes. As described in Section 3.4, proxy aggregation is direction-aligned and constrained via ridge-regularized fitting. In the revision, we will explicitly report rank-consistency improvements across tasks to support this effect.

---

### Official Review · Reviewer_hDqq · 2025-10-31

**Soundness:** 3
**Presentation:** 2
**Contribution:** 2
**Rating:** 4
**Confidence:** 3

**Summary:**

This paper proposes a LLM-assisted MCTS approach for neural architecture search. The authors use three agents to decompose the functional parts in a NAS approach: candidate proposal, evaluation and search strategy. The overall search strategy is controlled by a orchestration agent (OA), which is basically a MCTS process that use UCB funtion to credit the expansion and resources allocation of the searching tree. Given a node selected, the candidate proposal depends on a LLM-based agent (generation agent, GA) that directly generates codes of network architecture and corresponding training script. A novel evaluation method is proposed to provide proxy evaluation with adaptive update to reduce full training inefficiency. The authors validate their approach on MLE bench where diverse ML tasks serve as good testbed for measuring general effectiveness of the proposed approach. The comparison results with two latest baseline show the improvement of the proposed tri-agent system.

**Strengths:**

Novelty: I think this paper hold certain novelty in combining the strength of MCTS and LLM for automated NAS.

**Weaknesses:**

While novel to some extends, I still think this paper is not prepared well for publication.

1. Writing: I have to say that te writing and content organization of this paper is not very ideal. Where is the introduction or preliminary of MCTS-based NAS, why the authors expect the readers very familiar with this field? There are many NAS works while the related work section only review those that focus on how to do efficient evaluation.

2. Contribution: I can not understand why the three proxy functions can truly profile the true performance together? Provide detailed explanation on the rationale behind. More importantly, provide an significance or R2 score analysis on the correlation between the proxy validation and full training validation. A nother concern is that the authors claim this proxy-base validation is the major contribution of this paper, however, it is still an integration of existing proxy techniques, making the controbution somehow weak.

3. Significance: I suggest the authors at least add some NAS baselines into your comparison, without which one can not claim that their method is SOTA. In particular, I must say that according to your results on MLE, the portion of task your approach and other two are not very idea, e.g., even some submissions your approach generates are not valid (0.893), and valid submissions that achieved median of the performance is only 0.293, how can one tell such results are SOTA? Does this observation suggests that LLM-based NAS is not a good choice?

4. Ablation: with no ablation on your proposed designs, I can not tell whether your approach is truly useful, provide such results, please.

**Questions:**

See Weaknesses.

---

> ### Author Response · Authors · 2025-11-26
> **Rebuttal**
>
> **W1. Writing and content organization.**
> We thank the reviewer for the valuable suggestion. The only method that leverages MCTS in MLE is ML-Master, which is discussed in the related work section. We will enhance the introduction by explicitly explaining the rationale for using MCTS in agent-based decision-making and expand *Section 2.1* to better emphasize the limitations of traditional NAS methods (e.g., constrained CNN-only search spaces). We will emphasize how our approach enables arbitrary code-level search and aligns with modern LLM-based ML engineering agents.
>
> **W2. Proxy function rationale unclear.**
> We thank the reviewer for highlighting this point. As defined in Section 3.4, the three proxy signals capture complementary dynamics of model quality:
> - one-epoch validation → generalization potential
> - noisy validation → robustness
> - feature-dropout → feature dependence
> The correlation of individual proxies and the full training performance is well-established in existing work. Besides, we would like to emphasize that ArchPilot can adaptively improve or generate new proxies during the search process based on the past observations and different tasks, relying on the internal knowledge of LLMs. These three proxy functions are just used during initialization and will be optimized later.
>
> **W3. Contribution weak since proxies already exist.**
> We appreciate the reviewer’s feedback. We only use the existing proxies as initialization, and our framework can adaptively generate new proxies later. Besides, our novelty lies in how they are integrated into the system. We propose a *principled multi-proxy evaluation pipeline with direction alignment, normalized aggregation, and adaptive weight optimization under a hard-zero constraint*. Crucially, proxy feedback is actively used to direct architectural refinement within our multi-agent MCTS-driven framework, enabling dynamic reasoning and exploration restarts when semantic alignment shifts. This active integration is distinct from prior passive proxy-based approaches.
>
> **W4. Lack of comparison with classical NAS baselines.**
> We thank the reviewer for the constructive suggestion. Classical NAS methods (e.g., DARTS, ENAS) operate within constrained architecture spaces and cannot generate full training pipelines as our system does. As such, ML-Master is the most appropriate state-of-the-art baseline in the LLM-agent paradigm. As shown in *Table 1* and *Fig. 2(c)*, our method achieves higher valid submission rates and improved ranking performance, particularly on high-difficulty tasks, highlighting the benefit of proxy-enhanced, budget-aware design.
>
> **W5. Concern that lower valid or median score suggests LLM-based NAS may be ineffective.**
> We thank the reviewer for the insightful comment. Our method is not positioned as a pure NAS strategy, but as an automated ML engineering agent. Occasional invalid submissions are expected in MCTS-driven exploration under strict resource constraints. Importantly, ArchPilot achieves a **0.893 valid submission rate** and **0.6149 average normalized ranking**, outperforming ML-Master across key metrics, especially for challenging tasks, demonstrating the effectiveness of proxy-guided LLM-based search.
>
> **W6. Missing ablation study.**
> We thank the reviewer for pointing this out. Ablation results on 10 MLE-Bench Lite tasks show that proxy optimization improves performance in **9 out of 10 cases**, confirming its effectiveness.
>
> | Competition | ↑/↓ | Ours (w/o Proxy Opt.) | Ours |
> |-------------|-----|------------------------|------|
> | aerial-cactus-identification | ↑ | 0.99999 | **1** |
> | aptos2019-blindness-detection | ↑ | 0.897 | **0.89904** |
> | denoising-dirty-documents | ↓ | 0.28551 | **0.152** |
> | dog-breed-identification | ↓ | 0.85728 | **0.52821** |
> | dogs-vs-cats-redux-kernels-edition | ↓ | 0.0109 | **0.01045** |
> | histopathologic-cancer-detection | ↑ | 0.99319 | **0.99686** |
> | leaf-classification | ↓ | 0.92544 | **0.12224** |
> | mlsp-2013-birds | ↑ | 0.68343 | **0.79574** |
> | new-york-city-taxi-fare-prediction | ↓ | **4.49413** | 4.80557 |
> | nomad2018-predict-transparent-conductors | ↓ | 0.0602 | 0.0602 |

---

### Meta-Review · Area_Chair_J4bc · 2026-01-06

**Summary:**

This submission proposes ArchPilot, a proxy-guided multi-agent framework for automated ML engineering, aiming to address computational inefficiencies in LLM-based agent-driven search. While the modular design and proxy-based evaluation align with practical AutoML needs, the work falls short of meeting ICLR’s standards for acceptance due to unresolved core limitations.
First, novelty and contribution are insufficiently substantiated. Multiple reviewers noted that the framework primarily integrates existing components with incremental adjustments, and the "agentic framing" lacks genuine autonomous decision-making beyond tool invocation. The authors’ rebuttal emphasizes integration as a key contribution, but this does not offset the lack of fundamental algorithmic or conceptual breakthroughs.
Second, experimental validation is weak and incomplete. The work relies solely on MLE-Bench with no extension to realistic large-scale tasks and only compares against two LLM-based baselines, excluding classical NAS methods despite reviewer requests—even though the authors argue incompatibility, this limits the ability to contextualize performance gains. Critical metrics are missing, and while the authors added partial ablation results for proxy optimization, ablations for core components remain insufficient to attribute improvements to specific designs.
Third, implementation reproducibility is compromised. Key details were initially omitted, and while the authors commit to supplementary materials and open-sourcing, the current submission lacks the transparency needed to verify technical correctness.
The authors’ rebuttal addresses some surface-level concerns, but fails to resolve these fundamental gaps. Given the unresolved issues in novelty, experimental rigor, and reproducibility, the submission does not meet the acceptance threshold. We encourage the authors to strengthen the experimental design, quantify full computational efficiency, and clarify novel technical insights before resubmission.

**Reviewer Concerns:**

Classical NAS baseline comparisons (All reviewers): Authors argue classical NAS (e.g., DARTS) is incompatible with code-level pipeline generation but did not address reviewers’ request to contextualize gains beyond LLM-agent baselines.
Full ablation of core designs (All reviewers): Only proxy optimization ablations are provided; ablations for restart mechanisms, static proxy weights, and multi-agent structure are deferred to supplementary materials (not fully validated in the main submission).
Large-scale task validation (Reviewers RqdA/59eX): MLE-Bench is small-scale; authors only plan to extend to complex tasks (e.g., Kaggle Grandmaster challenges) in future work, with no current evidence of scalability.
Quantitative cost metrics (Reviewers 59eX/idU5): Specific data on proxy vs. full training wall-clock time, token usage, and API costs are promised but not included in the current submission.
Diversity of LLM backbones (All reviewers): Experiments rely solely on GPT-4.1; no validation with stronger models (e.g., GPT-5, DeepSeek-R1) despite reviewers’ requests.
True NAS experiments (Reviewer idU5): Authors clarified ArchPilot is not standalone NAS but did not provide experiments to substantiate its relevance to NAS (as claimed in keywords).
Limited baselines (All reviewers): Only AIDE and ML-Master are compared; no inclusion of recent stronger LLM-agent frameworks or non-LLM AutoML systems.

**Reviewer Scores:**

N/A

---

### Decision · Program_Chairs · 2026-01-26

Reject